# Tegaserod Maleate Suppresses the Growth of Gastric Cancer In Vivo and In Vitro by Targeting MEK1/2

**DOI:** 10.3390/cancers14153592

**Published:** 2022-07-23

**Authors:** Zitong Wang, Yingying Chen, Xiaoyu Li, Yuhan Zhang, Xiaokun Zhao, Hao Zhou, Xuebo Lu, Lili Zhao, Qiang Yuan, Yunshu Shi, Jimin Zhao, Ziming Dong, Yanan Jiang, Kangdong Liu

**Affiliations:** 1Pathophysiology Department, School of Basic Medical Sciences, Zhengzhou University, Zhengzhou 450001, China; zitongwang1997@163.com (Z.W.); chenchenchengying@163.com (Y.C.); xiaoyuli520000@163.com (X.L.); yuhanz125@163.com (Y.Z.); zhaoxkdyx@163.com (X.Z.); hzzhou0317@gmail.com (H.Z.); solidlxb@hotmail.com (X.L.); lilizhao184@163.com (L.Z.); yuanqiang401@163.com (Q.Y.); sysnan@163.com (Y.S.); zhaojimin@zzu.edu.com (J.Z.); zlyyqp2406@zzu.edu.cn (Z.D.); 2China-US (Henan) Hormel Cancer Institute, Zhengzhou 450001, China; 3State Key Laboratory of Esophageal Cancer Prevention and Treatment, Zhengzhou 450001, China; 4Basic Medicine Research Center, Zhengzhou University, Zhengzhou 450001, China; 5Provincial Cooperative Innovation Center for Cancer Chemoprevention, Zhengzhou 450001, China; 6Cancer Chemoprevention International Collaboration Laboratory, Zhengzhou 450001, China

**Keywords:** gastric cancer, tegaserod maleate, MEK1, MEK2, chemoprevention

## Abstract

**Simple Summary:**

The MAP kinase cascades are the most important oncogenic drivers of human cancers, including gastric cancer, and blocking this pathway with targeted inhibitors is an important antitumor strategy. Tegaserod maleate binds to MEK1 and MEK2, inhibiting kinase activity and thus suppressing the MEK1/2-ERK1/2 signaling pathway. Thus, tegaserod maleate may have potential as a MEK1/2 inhibitor for gastric cancer.

**Abstract:**

Gastric cancer (GC) ranks fifth in global incidence and fourth in mortality. The current treatments for GC include surgery, chemotherapy and radiotherapy. Although treatment strategies for GC have been improved over the last decade, the overall five-year survival rate remains less than 30%. Therefore, there is an urgent need to find novel therapeutic or preventive strategies to increase GC patient survival rates. In the current study, we found that tegaserod maleate, an FDA-approved drug, inhibited the proliferation of gastric cancer cells, bound to MEK1/2 and suppressed MEK1/2 kinase activity. Moreover, tegaserod maleate inhibited the progress of gastric cancer by depending on MEK1/2. Notably, we found that tegaserod maleate suppressed tumor growth in the patient-derived gastric xenograft (PDX) model. We further compared the effect between tegaserod maleate and trametinib, which is a clinical MEK1/2 inhibitor, and confirmed that tegaserod maleate has the same effect as trametinib in inhibiting the growth of GC. Our findings suggest that tegaserod maleate inhibited GC proliferation by targeting MEK1/2.

## 1. Introduction

Gastric cancer (GC) is characterized by insidious onset, asymptomatic or minor symptoms at the early stage, causing delayed diagnosis and poor survival for most patients [1,2,3]. At present, radical gastrectomy is the only potentially curative treatment for GC. Despite curative resection, recurrences occur in about 60% of patients. The main reason is that GC is usually diagnosed at the advanced stage [4,5]. Preventing or reducing the frequency of the recurrence is probably more crucial than early detection of recurrence [6,7]. Hence, the development of novel drugs with low toxicity and high efficiency to prevent GC recurrence is a promising strategy to improve the survival rate of patients.

Patients with GC benefit from adjuvant therapy to reduce recurrence after curative resection [8]. However, although postoperative chemoradiotherapy has an important effect on locoregional recurrence in patients with operable gastric cancer, there appears to be no additional benefit for regional and distant recurrence [9]. Chemoprevention is involved in the use of natural or synthetic chemical agents to reverse, suppress or prevent the progression of a potentially malignant tumor to aggressive cancer and prevent cancer recurrence, thereby reducing cancer morbidity and mortality [3,10,11]. It is reported that celecoxib, a COX-2 inhibitor, has a significant effect on the regression of precancerous gastric lesions [12]. Even with these encouraging results, the evidence for chemoprevention of gastric cancer recurrence is limited [13]. 

The development and approval of new drugs is a difficult and costly task with a high failure rate [14,15]. Repurposing is an attractive drug development strategy since the approved drugs have known pharmacokinetic characteristics, dosage, safety, and efficacy [16,17]. Therefore, it is of great significance to screen the FDA-approved drug library to select drugs with low toxicity and high efficiency to prevent cancer or recurrence [18]. In recent years, it has been reported that tegaserod maleate, a 5-hydroxytryptamine 4 receptor (5-HT4R) agonist [19], can inhibit the proliferation of cancer cells such as esophageal squamous cell carcinoma, melanoma, prostatic cancer and lung cancer [20,21]. However, the inhibitory effect of tegaserod maleate on gastric cancer and its underlying mechanism remains unclear.

Here, we found that tegaserod maleate significantly suppressed the proliferation of GC in vitro and in vivo by inhibiting MEK1/2, and provided reasonable theoretical support for the application of tegaserod maleate in the chemoprevention of gastric cancer.

## 2. Materials and Methods

### 2.1. Reagents and Antibodies

Tegaserod maleate (CAS:189188-57-6) was purchased from Selleck Chemicals LLC (Houston, TX, USA). Tegaserod maleate was dissolved with dimethyl sulfoxide to the final concentration 50 mM and stored at −80 °C. The following antibodies were used in the study: anti-phospho-ERK1/2 (Thr202/Tyr204) (Cat# 4370, Cell Signaling Technology, Danvers, MA, USA), anti-ERK1/2 (Cat#4695, Cell Signaling Technology), anti-MEK1 (Cat#2352, Cell Signaling Technology), anti-MEK2 (Cat#9125, Cell Signaling Technology), anti-Ki-67 antibody (Cat#ab15580, Abcam, Cambridge, UK), anti-Flag (Cat #F1804, Sigma, St. Louis, MO, USA).

### 2.2. Cell Culture

Human GC cell lines HGC27 and AGS were obtained from the Chinese Academy of Sciences (Beijing, China). HGC27 cell line was cultured in RPMI-1640 medium containing 10% fetal bovine serum at 37 °C in a 5% CO_2_ humidified incubator. AGS cell line was maintained in F12K medium supplemented with 10% fetal bovine serum and cultured at 37 °C in a 5% CO_2_ humidified incubator.

### 2.3. Cell Proliferation Assay

GC cells (HGC27 cells: 2 × 10^3^ cells/well; AGS cells: 3 × 10^3^ cells/well) were seeded in 96-well plates. After incubation in the incubator for 16–18 h, the GC cells were treated with tegaserod maleate (0, 0.25, 0.5, 1 and 2 μM) for 24 h, 48 h, 72 h and 96 h. MTT (1 mg/mL) was added at a ratio of 1:100. After incubation for 2 h, 100 μL DMSO was added to terminate the reaction, and OD value at 490 nm/570 nm was measured with a microplate analyzer.

### 2.4. Anchorage-Independent Cell Growth Assay

After the base layer agar was prepared with 0, 0.25, 0.5, 1 or 2 μM tegaserod maleate, HGC27 and AGS cells (8 × 10^3^ cells/well) were seeded in top layer agar with 0, 0.25, 0.5, 1 or 2 μM tegaserod maleate. The cells were cultured at 37 °C in a 5% CO_2_ incubator for 14 days. The colonies were counted using IN Cell Analyzer 6000 software. 

### 2.5. Anchorage-Dependent Cell Growth Assay

After seeding HGC27 (500 cells/well) and AGS (500 cells/well) in 6-well plates, cells were treated with 0, 0.25, 0.5, 1 or 2 μM tegaserod maleate. Two weeks later, the cells were fixed with 4% paraformaldehyde at 25 °C for 30 min and stained with 0.1% crystal violet for 3 min.

### 2.6. Western Blotting

The HGC27 and AGS cells were inoculated into 10 cm dish. After 16 h incubation, tegaserod maleate with different concentrations was added and treated cells for 24 h. The cells were harvested and lysed using RIPA lysis buffer to obtain protein samples. Protein concentration was detected using the bicinchoninic acid protein assay kit (BCA Protein Assay Kit, Beyotime Biotechnology, Shanghai, China). Next, 30 μg protein extract was separated using sodium dodecyl sulfate-PAGE and transferred to polyvinylidene fluoride (PVDF) membranes. The membrane was blocked with 5% skim milk for 1 h at room temperature. Next, the membranes were incubated overnight with the primary antibody at 4 °C and then for 2 h with the secondary antibody at room temperature. Protein bands were displayed using the enhanced chemiluminescence (ECL) detection reagent (Dalian Meilun Biotechnology Co., Ltd., Dalian, China). Original Western Blotting figures shown in Appendix A. 

### 2.7. Pull-Down Assay

Epoxy-activated sepharose 4B conjugated with tegaserod maleate was prepared according to the manufacturer’s instructions (#17-0480-01, GE Healthcare Life Sciences, Uppsala, Sweden). An amount of 200 μL tegaserod maleate-Sepharose 4B beads or DMSO were rotated with cell proteins (500 μg) in the reaction buffer. After rocking gently for 48 h at 4 °C, the conjugated beads and DMSO were washed 4 times in the washing buffer followed by the addition of 30 μL 3× loading buffer at 100 °C for 5 min. The binding was detected using Western blotting.

### 2.8. In Vitro Kinase Assay

Inactive ERK2 protein (50 ng) was incubated with active MEK1 recombinant protein (200 ng) or MEK2 recombinant protein (300 ng) for in vitro kinase assay. The reaction was run in kinase buffer (Cat#K02–09, SignalChem, Canada) for 30 min at 30 °C and stopped using 6 × loading buffer. The protein was assessed using Western blotting.

### 2.9. Cellular Thermal Shift Assay (CETSA)

MEK1/2 protein was overexpressed in 293F cells. After 16 h incubation, tegaserod maleate was added and treated cells for 24 h. For the cell lysate CETSA, cultured cells were harvested and washed with PBS. The respective lysates were divided into smaller (100 μL) aliquots and heated at different temperatures for 3 min followed by cooling at room temperature for 3 min. The cell suspensions were frozen and thawed twice with liquid nitrogen. The lysate was separated from the cell debris using centrifugation at 12,000× *g* for 20 min at 4 °C. The supernatants were transferred to the new microtubes and analyzed using Western blotting.

### 2.10. Computer Docking Model

The Schrödinger Suite 2015 software program was used for silico docking. The three-dimensional (3D) structure of tegaserod maleate was derived from PubChem Compound. The Protein Preparation Wizard standard pipeline (Schrödinger Suite 2015) was used to process the structure of MEK1 (PBD:3EQG) and MEK2 (PBD:1S9I).

### 2.11. CRISPR/Cas9 Knockout Cell Lines

MEK1/2 were deleted in GC cells using the CRISPR/Cas9 system according to the manufacturer’s protocols. Viral plasmids and packaging vectors (pMD2.G and psPAX2) were transfected into 293T cells using Jet Primer (ThermoFisher Scientific, Waltham, MA, USA). After 4 h transfection, the medium was removed and cells were cultured for 24 h, 48 h and 72 h. Viral particles were harvested and filtered using a 0.22 μm syringe filter. HGC27 and AGS cells were infected with viral mixture (3 mL viral particles, 7 mL complete medium and 8 μg/mL polybrene). The cells were treated with 2 μg/mL and 1 μg/mL puromycin for 72 h, respectively. The knockout efficacy was detected using Western blotting. The oligonucleotide sequences of MEK1 and MEK2 single guide (sg) RNA are listed in Table 1.

### 2.12. PDX Model

Severe combined immunodeficiency (SCID) mice were used (6 to 8 weeks old; Vital River Labs, Beijing, China) to investigate the effect of tegaserod maleate on GC PDX tumor growth, GC tissues were cut into about 1–2 mm pieces and planted into the back of the mice. The mice were divided into three groups: vehicle group, low-dose group (2 mg/kg) and high-dose group (10 mg/kg), with ten mice in each group. The vehicle or tegaserod maleate was taken orally once daily. Each mouse was checked twice weekly for tumor volume and body weight. The tumor was extracted until the tumor volume reached 1000 mm^3^, and then the tumor weight was measured.

### 2.13. Immunohistochemistry (IHC) Analysis

The tumor tissues were extracted and embedded in paraffin blocks for IHC staining. The tissue sections were deparaffinized, hydrated, antigen-repaired and blocked. Sections were incubated at 4 °C with primary antibodies of Ki-67 overnight and then for 15 min with the secondary antibody at 37 °C, followed by diaminobenzidine and hematoxylin staining. Then, the sections were dehydrated and covered with slides. All tissue sections were photographed with a microscope camera and analyzed using TissueFAXS Viewer software.

### 2.14. Statistical Analysis

SPSS statistical software, version 21 (IBM Corp.), was used for all statistical analyses. The results used non-parametric test or one-way analysis of variance (ANOVA) to compare significant differences. Quantitative data were indicated as mean values ± standard deviations. Statistical significance was defined by a *p*-Value < 0.05.

## 3. Results

### 3.1. Tegaserod Maleate Inhibits GC Cell Proliferation

Tegaserod maleate was used for the treatment of irritable bowel syndrome in clinic. The chemical structure of tegaserod maleate is shown in Figure 1A. We first determined the toxicity of tegaserod maleate in GC cells using an MTT assay, and found that the half-maximal inhibitory concentration (IC50) values were 1.40 μM and 2.14 μM for HGC27 and AGS cells at 48 h, respectively (Figure 1B). Hereafter, we selected 0.25, 0.5, 1 and 2 μM tegaserod maleate for further studies. The results revealed that tegaserod maleate (2 μM) inhibited HGC27 cell proliferation by 31.93% and 38.38% at 72 h and 96 h, respectively, and AGS cell proliferation by 36.29% and 40.69% at 72 h and 96 h, respectively (Figure 1C). Then, we confirmed the anticancer effects of tegaserod maleate using an anchorage-independent and anchorage-dependent cell growth assay. Compared to the control group, the results showed that the colony numbers of HGC27 and AGS cells both decreased in a dose-dependent manner after tegaserod maleate treatment (Figure 1D,E and Appendix A). Hence, these results confirmed that tegaserod maleate could inhibit GC cell proliferation in vitro.

### 3.2. Tegaserod Maleate Binds to MEK1 and MEK2

Since tegaserod maleate inhibited GC cell proliferation, we screened its target using an in-silico docking assay. Our results predicted that tegaserod maleate achieved binding with MEK1 at ASP208, MET146 and ASP152 sites, and with MEK2 at GLY83 and GLY81 sites (Figure 2A). Then, the computational docking results were verified using a pull-down assay. We firstly attested that tegaserod maleate could bind to recombinant MEK1 and MEK2 proteins ex vitro (Figure 2B and Appendix A). Then, we overexpressed MEK1 and MEK2 proteins in HEK293F cells and found that tegaserod maleate could also bind to MEK1 and MEK2 proteins of HEK293F (Figure 2C and Appendix A). We then tested whether tegaserod maleate could bind to MEK1 and MEK2 kinases in HGC27 lysate or AGS lysate. The results exhibited that tegaserod maleate could bind to endogenous MEK1 and MEK2 proteins (Figure 2D and Appendix A). Similarly, CETSA results showed that the Tm values of the control group were 51.1 °C and 52.6 °C, while the Tm values of the tegaserod maleate treatment group were 60.3 °C and 57.4 °C. The results indicated that the Tm values shifted to the right and further verified that the MEK1 and MEK2 proteins in intact cells were more stable after tegaserod maleate treatment (Figure 2E, Appendix A). We then examined whether ASP208, MET146 and ASP152 of MEK1 and GLY83 and GLY81 of MEK2 bound with tegaserod maleate by mutating these sites. After mutation, the cell lysate was harvested for pull-down assay, and the results showed that the binding ability of tegaserod maleate to these mutational proteins was decreased (Figure 2F and Appendix A). These results suggested that tegaserod maleate directly bound to MEK1/2.

### 3.3. Tegaserod Maleate Inhibits the MEK1/2-ERK1/2 Signaling Pathway in GC

Our above data indicated that tegaserod maleate could bind with MEK1 and MEK2. Then, we investigated whether tegaserod maleate inhibited the kinase activity of MEK1/2 or not. We found that the phosphorylation efficiency of active MEK1 and MEK2 against inactive ERK2 was decreased at 0.25, 0.5, 1 and 2 μM tegaserod maleate (Figure 3A and Appendix A). These results suggested that tegaserod maleate could cause a decrease in MEK1/2 kinase activity and directly lead to suppressing ERK2 activation. Furthermore, we evaluated the change of downstream molecules of MEK1 and MEK2. The results indicated that tegaserod maleate suppressed the phosphorylation of ERK1/2 in HGC27 and AGS cells. RSK2, the downstream molecule of ERK, is a key signaling molecule involved in cell proliferation and cancer development [22]. After tegaserod maleate treatment, the levels of RSK2 S227 decreased in a dose-dependent pattern (Figure 3B and Appendix A). Next, in order to verify that tegaserod maleate inhibited the activation of ERK via MEK1 and MEK2 in cells, we performed an immunoprecipitation assay and the results showed that MEK1 and MEK2 could combine with ERK1/2 and tegaserod maleate could decrease the level of p-ERK1/2 T202/Y204 through suppressing the MEK1 and MEK2 kinase activity (Figure 3C,D and Appendix A). In addition, the immunofluorescence assay was performed to investigate the variation of ERK1/2 T202/Y204 phosphorylation. The results showed that tegaserod maleate inhibited the fluorescence intensity of p-ERK1/2 T202/Y204 in a dose-dependent manner (Figure 3E and Appendix A). Therefore, the above data supported that tegaserod maleate suppressed cell proliferation of GC by blocking the MEK1/2-ERK1/2 pathway.

### 3.4. Tegaserod Maleate Inhibits Gastric Cancer Cell Growth by Depending on MEK1/2

To evaluate the role of MEK1 and MEK2 in GC progression, AGS and HGC27 cell lines were used for the knockout assay. The results indicated that sgMEK1#2 and sgMEK1#4 reduced the protein levels of MEK1 significantly in both HGC27 and AGS gastric cell lines, while sgMEK2#4 and sgMEK2#5 could reduce the protein level of MEK2 (Figure 4A and Appendix A). After knocking out of MEK1, the proliferation inhibition rates were 68.91% and 80.97% for HGC27 cells, and 65.51% and 84.09% for AGS cells, respectively. Similarly, after MEK2 deletion, the proliferation inhibition rates were 82.05% and 47.70% for HGC27 cells, and 85.78% and 53.30% for AGS cells, respectively. These results showed that the proliferation of HGC27 and AGS was suppressed (Figure 4B). Similarly, the anchor-dependent cell growth was also decreased in the knockout groups (Figure 4C and Appendix A). Our data showed that tegaserod maleate could effectively inhibit GC cell proliferation and colony formation. However, it is unclear whether the inhibitory effect of tegaserod maleate depends on levels of MEK1 and MEK2 or not. Therefore, we treated MEK1 and MEK2 knockout GC cells with tegaserod maleate. After tegaserod maleate treatment for 96 h, cell viability was evaluated in HGC27 and AGS cell lines in control and knockout cell groups. The results showed that the GC cell viability with tegaserod maleate treatment in the sgcontrol group was lower than those of the sgMEK1#2 and sgMEK1#4 groups. Similarly, compared with the sgcontrol group, the cell viability of sgMEK2#4 and sgMEK2#5 was increased after tegaserod maleate treatment (Figure 4D). These results suggested that tegaserod maleate indeed exerted an inhibitory effect that was dependent on MEK1 and MEK2.

### 3.5. Tegaserod Maleate Inhibits the Growth of GC PDX Models

In order to evaluate the antitumor activity of tegaserod maleate in vivo, PDX models of LSG85 and LSG51 cases with high MEK1 levels were established from the GC PDX specimen repository (Figure 5A,B and Appendix A). After tegaserod maleate was administered to the mice daily by gavage, the tumor volumes of the high-dose tegaserod maleate treatment group both in LSG85 and LSG51 cases were inhibited when compared with the vehicle group (Figure 5C). After sacrificing the mice and removing the tumors, the tumor pictures in both LSG85 and LSG51 cases were shown in Figure 5D. We assessed tumor weight and detected that tumor growth inhibition (TGI) reached 34.32% in the LSG85 case and 56.70% in the LSG51 case with 10 mg/kg tegaserod maleate (Figure 5E). In addition, immunohistochemical staining showed that the positive rates of the Ki-67 were reduced after tegaserod maleate treatment in LSG85 and LSG51 cases (Figure 5F). Synchronously, we investigated the levels of p-ERK1/2 T202/Y204 in tumor tissues using Western blotting. These results showed that, compared with the vehicle group, the levels of p-ERK1/2 T202/Y204 were inhibited in the tegaserod maleate-treatment group (Figure 5G and Appendix A). In conclusion, tegaserod maleate suppressed GC tumor growth by inhibiting the MEK1/2-ERK1/2 signaling pathway in vivo.

### 3.6. Tegaserod Maleate Has the Same Inhibitory Effect Compared with MEK1/2 Inhibitor Trametinib

To confirm whether tegaserod maleate can be used as an inhibitor of MEK1 and MEK2 in gastric cancer, we compared the anti-tumor effects of tegaserod maleate with trametinib, an FDA-approved MEK1/2 inhibitor. The dosages of trametinib were selected based on previous studies [23]. We selected the HSG288 GC PDX model for further study (Figure 6A). The tumor volumes of both the tegaserod maleate groups and the trametinib group in HSG288 cases were also decreased (Figure 6B). After sacrificing the mice and removing the tumors, the tumor pictures were shown in Figure 6C. Subsequently, we assessed tumor weight and detected that TGI reached 68.41% with 10 mg/kg tegaserod maleate. Importantly, there was no statistically significant difference between the tegaserod maleate-treated group and the trametinib-treated group (Figure 6D). Furthermore, we examined the effect of tegaserod maleate on Ki-67 expression by IHC. The quantified IHC results were more likely to exhibit that the tegaserod maleate-treated groups decreased protein levels of Ki-67 (Figure 6E and Appendix A). Then, in order to verify the toxic effect of tegaserod maleate, we performed HE staining on the heart, liver, spleen, lung and kidneys of the mice, and the results showed that there was no toxic effect on these organs (Appendix A). In conclusion, tegaserod maleate had the potential to be applied in GC chemoprevention as a MEK1/2 inhibitor.

## 4. Discussion

Despite improvements in adjuvant therapies and general progress in oncogenesis mechanism comprehension for gastric cancer in recent years, the long-term survival of gastric cancer patients is not substantially improved due to high recurrence rates and a dismal prognosis [24,25]. As a result, chemoprevention strategies still appear an effective strategy in reducing the incidence and mortality of GC cancer [26,27]. Accumulating evidence demonstrates that drugs targeting nutrient metabolism are promising candidates for preventing tumorigenesis and cancer recurrence because of their availability and relatively low side effects compared to other chemotherapeutic drugs [28]. Notably, many FDA-approved drugs that target nutrient metabolism, such as metformin, thiazolidinediones, statins, specific amino acid deprivation drugs and drugs targeting protein degradation, have shown significant antitumor effects in various types of cancer [28,29,30,31]. Moreover, by screening the FDA-approved drug library, our previous study confirmed that tegaserod maleate suppresses the proliferation of ESCC by blocking peroxisome function and the features of tegaserod maleate make it a promising candidate for cancer chemoprevention [32]. However, the underlying molecular targets and related mechanisms still need be further illustrated. In our study, we identified that tegaserod maleate also exhibited anticancer effects on GC cells. Moreover, we confirmed that tegaserod maleate had an obvious inhibitory effect on the gastric cancer PDX model and had the same inhibitory effect in inhibiting tumor volume or tumor weight as trametinib. According to clinical studies, trametinib, an FDA-approved MEK1/2 inhibitor, was used in tumors with BRAF-activating mutations including melanoma, non-small cell carcinoma and thyroid cancer [33,34]. However, trametinib has not been used in the clinical treatment of gastric cancer yet [35]. Consequently, our findings support that tegaserod maleate could be used in GC chemoprevention as a MEK1/2 inhibitor.

The MAP kinase cascade is the most important oncogenic driver of human cancers, and blocking this signaling module by targeted inhibitors is a promising anti-tumor strategy [36,37]. Since MEK1/2 has very narrow substrate specificity, MEK1/2 inhibition specifically shuts off ERK1/2 signaling without affecting other signaling pathways [38]. Therefore, MEK1/2 is an efficient target for anticancer therapy and inhibition of MEK1/2 is an effective method to suppress the whole cascade reaction [39]. Furthermore, we detected the role of MEK1 and MEK2 in GC growth and found that the MEK1 and MEK2 protein levels were higher in GC (Appendix A). In addition, the protein levels of MEK1 and MEK2 were negatively correlated with the survival rate of GC patients (Appendix A). Combined with the MEK1 and MEK2 knockout assays in GC cells, the results identified that tegaserod maleate inhibited GC proliferation through MEK1 and MEK2. In addition, we utilized a computational docking model to indicate that tegaserod maleate could bind to MEK1 and MEK2 (Figure 2). Then, we also verified it using a pull-down assay and CETSA assay (Figure 2). The homology of the full-length protein sequence between MEK1 (45 kDa) and MEK2 (46 kDa) was 85%, and the homology of the catalytic domain was 86%. The typical MEK1/2 secondary structures consist of the N-terminal ~70 amino acid residues, the protein kinase domain (~290 amino acids), and the C-terminal ~30 amino acid residues [35,40]. The ASP208, MET146 and ASP152 were located in the kinase domain of MEK1, and GLY81 and GLY83 were located in the kinase domain of MEK2. Our results indicated that tegaserod maleate could bind with MEK1 at ASP208, MET146 and ASP152 and MEK2 at GLY81 and GLY83, respectively (Figure 2). Furthermore, we confirmed that tegaserod maleate suppressed the MEK1 and MEK2 kinase activity using in vitro kinase assay. These data indicate that tegaserod maleate is a novel inhibitor of MEK1/2.

In conclusion, our findings attested that MEK1 and MEK2 are promising therapeutic targets for GC. Tegaserod maleate could bind to MEK1 and MEK2 and inhibit its kinase activity. Tegaserod maleate also suppressed GC growth in vitro and in vivo by blocking the MEK1/2-ERK1/2 signaling pathway. Our study suggests that the proper application of tegaserod maleate may be a beneficial therapeutic strategy for GC patients with high MEK1/2 levels.

## 5. Conclusions

Tegaserod maleate binds with MEK1 and MEK2 and inhibits kinase activity, thereby blocking the MEK1/2-ERK1/2 signaling pathway to inhibit the proliferation of gastric cancer cells in vitro and in vivo.

## Figures and Tables

**Figure 1 cancers-14-03592-f001:**
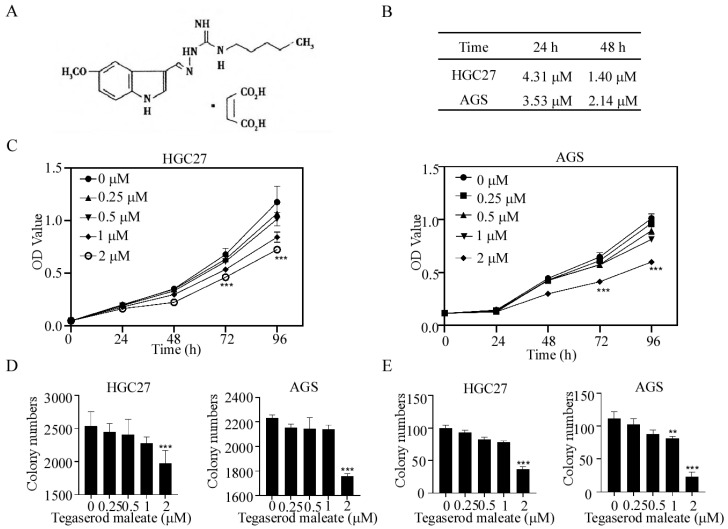
Tegaserod maleate inhibits GC cell proliferation. (**A**) Chemical structure of tegaserod maleate. (**B**) Cell toxicity assay. AGS and HGC27 treated with tegaserod maleate (0, 3.125, 6.25, 12.5, 25, 50 μM) for 24 and 48 h were evaluated using MTT assay. (**C**) Tegaserod maleate has an inhibitory effect on gastric cancer cells. AGS and HGC27 treated with tegaserod maleate (0, 0.25, 0.5, 1, 2 μM) for 0, 24, 48, 72 and 96 h were evaluated using MTT assay. (**D**) Tegaserod maleate inhibited anchorage-independent gastric cancer cell growth. AGS and HGC27 cells treated with tegaserod maleate (0, 0.25, 0.5, 1, 2 μM) for 2 weeks. (**E**) Tegaserod maleate inhibited anchorage-dependent gastric cancer cell growth. AGS and HGC27 cells treated with tegaserod maleate (0, 0.25, 0.5, 1, 2 μM) for 2 weeks. The asterisks (**) (***) indicate a significant (*p* < 0.01 and 0.001).

**Figure 2 cancers-14-03592-f002:**
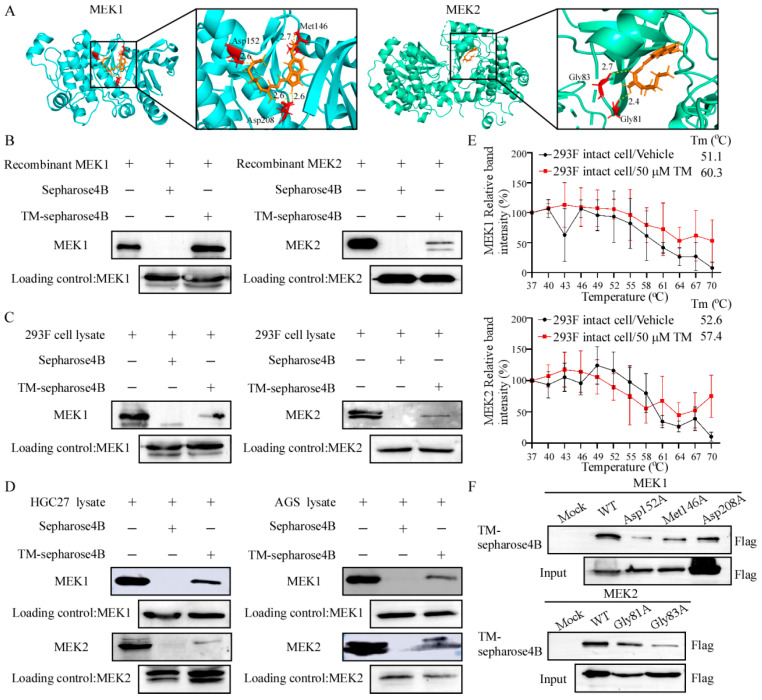
Tegaserod maleate binds to MEK1/2. (**A**) Computational docking model between tegaserod maleate and MEK1 or MEK2. The binding ability of tegaserod maleate to recombinant MEK1 and MEK2 protein (**B**), overexpressed MEK1 and MEK2 protein in 293F cells (**C**) and endogenic MEK1 and MEK2 protein in vitro (**D**), obtained via pull-down assay. (**E**) Cellular thermal shift assay. The protein stability of MEK1 and MEK2 in intact cells. (**F**) The binding ability of tegaserod maleate to mutant MEK1 and MEK2.

**Figure 3 cancers-14-03592-f003:**
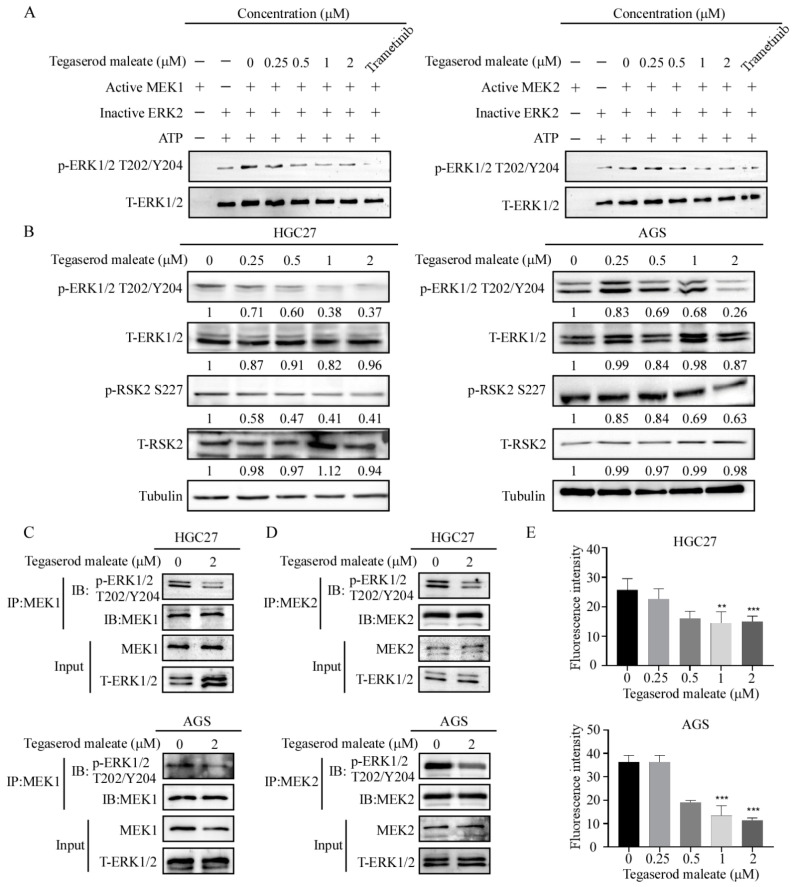
Tegaserod maleate inhibits the MEK1/2-ERK1/2 signaling pathway in GC. (**A**) MEK1 and MEK2 kinase activity was assessed by in vitro kinase assay using active MEK1, MEK2 and inactive ERK2 proteins. The effect of tegaserod maleate was determined using Western blotting. (**B**) The levels of p-ERK1/2, ERK1/2, p-RSK2 and T-RSK2 in HGC27 and AGS cells with different concentrations of tegaserod maleate (0, 0.25, 0.5, 1 and 2 μM) treatment for 24 h was determined by Western blotting. (**C**) The levels of p-ERK1/2 T202/Y204 were affected by MEK1 and MEK2 in HGC27 (**C**) and AGS (**D**) when treated with tegaserod maleate. ERK1/2 was immunoprecipitated by MEK1/2 and ERK1/2 was detected using p-ERK1/2 T202/Y204 (**E**) Immunofluorescence staining of HGC27 and AGS: cells were treated for 24 h, and then stained for p-ERK1/2 T202/Y204 (100 magnifications). The asterisks (**) (***) indicate a significant (*p* < 0.01 and 0.001).

**Figure 4 cancers-14-03592-f004:**
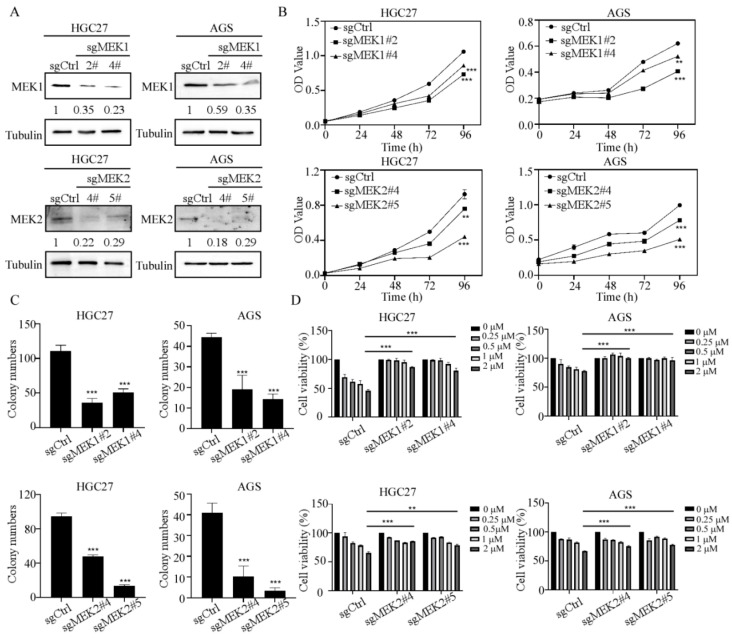
Tegaserod maleate inhibits gastric cell growth by depending on MEK1/2. (**A**) MEK1 and MEK2 knockout efficiency was assessed in HGC27 and AGS cells. (**B**) Sgcontrol and sgMEK1 or sgMEK2 groups were detected for 0, 24, 48, 72 and 96 h. Cell proliferation was evaluated using MTT assay. (**C**) Anchoring dependence ability of sgcontrol and sgMEK1 or sgMEK2 groups. (**D**) Sgcontrol and sgMEK1 or sgMEK2 groups were treated with either tegaserod maleate or DMSO for 96 h. Cell viability was evaluated using MTT assay. The asterisks (**) (***) indicate a significant (*p* < 0.01 and 0.001).

**Figure 5 cancers-14-03592-f005:**
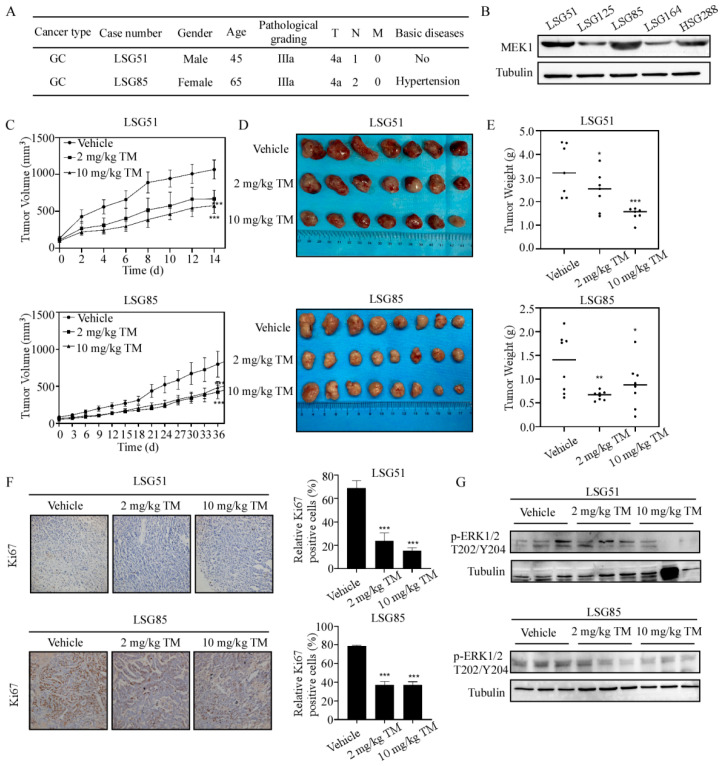
Tegaserod maleate inhibits the growth of GC PDX. (**A**) Patients’ information for samples used in the GC PDX models. (**B**) The protein levels of MEK1 in different PDX cases. (**C**) The change of average tumor volume in different groups of LSG85 and LSG51 cases after tegaserod maleate treatment. (**D**) Tumor images of different groups after sacrifice. LSG85 (n = 8), LSG51 (n = 7). (**E**) Tumor weight analysis in different groups of LSG85 and LSG51 cases after tegaserod maleate treatment compared with the average tumor weight of the vehicle group. (**F**) Left panel: Representative IHC images of LSG85 and LSG51 tumor tissue slices (100 magnifications), tumor tissues were stained with Ki-67 antibody; Right panel: Statistical analysis of IHC positive staining of Ki-67 in both LSG85 and LSG51 cases. (**G**) The protein levels of p-ERK1/2T202/Y204 in tumor tissues were detected using Western blotting. The asterisks (*) (**) (***) indicate a significant (*p* < 0.05, 0.01 and 0.001).

**Figure 6 cancers-14-03592-f006:**
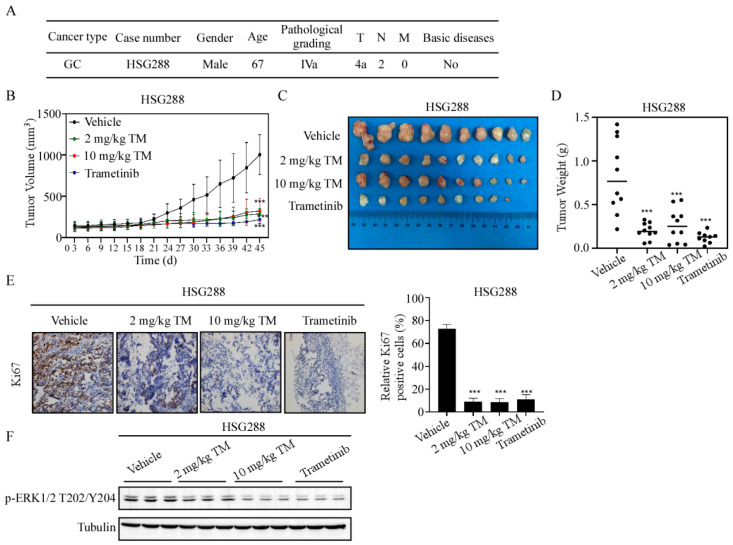
Tegaserod maleate has the same inhibitory effect compared with MEK1/2 inhibitor trametinib. (**A**) Patient information for samples used in the GC PDX models. (**B**) The change in average tumor volume in different groups of HSG288 cases after tegaserod maleate and trametinib treatment. (**C**) Tumor images of different groups after sacrifice. HSG288 (n = 10). (**D**) Tumor weight analysis in different groups of HSG288 cases after tegaserod maleate and trametinib treatment. (**E**) Left panel: Representative IHC images of HSG288 tumor tissue slices (100 magnifications), tumor tissues were stained with Ki-67 antibody; Right panel: Statistical analysis of IHC positive staining of Ki-67 in HSG288 cases. (**F**) The protein levels of p-ERK1/2T202/Y204 in tumor tissues were detected using Western blotting. The asterisks (***) indicate a significant (*p* < 0.001).

**Table 1 cancers-14-03592-t001:** The oligonucleotide sequences of MEK1 and MEK2 single guide (sg) RNA.

Gene Name	Primer Sequences 5′-3′
sgMEK1#2	F:CGTTAACTGCAGAGCCGTCG
	R:CGACGGCTCTGCAGTTAACGC
sgMEK1#4	F:GCAGCAGCGAAAGCGCCTTG
	R:CAAGGCGCTTTCGCTGCTGC
sgMEK2#4	F:GACGGCGAGTTGCATTCGTGCAGG
	R:CCTGCACGAATGCAACTCGCCGT
sgMEK2#5	F:GCACACATTACTCGGTGCAGTCGG
	R:CCGACTGCACCGAGTAATGTGTG

F = Forward, R = Reverse.

## Data Availability

The data generated and/or analyzed during the current study are available from the corresponding author on reasonable request.

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
