# Peer review of "Tegaserod Maleate Suppresses the Growth of Gastric Cancer In Vivo and In Vitro by Targeting MEK1/2"

_cancers, 2022, doi:10.3390/cancers14153592_

Round 1
Reviewer 1 Report
The authors demonstrate that tegaserod maleate, known as an agonist of 5-hydroxytryptamine 4 receptor, inhibits the proliferation of gastric cancer (GC) cells and suppresses tumor growth in patient-derived gastric xenograft models. With the use of GC cell lines, the authors showed that tegaserod maleate is capable of inhibiting GC proliferation through MEK1 and MEK2. The experimental work is well designed and the results are presented appropriately. The minor points indicating below must be fixed.
Figure 3c
HGC27, IB pERK1/2T202/Y204: the quality of immunoblot is low, and bands are not clearly evident. Same situation is observed for the input MEK1.
Figure 3d
HGC27, IB pERK1/2T202/Y204: there is not a clear difference in band intensity between 0 and 2 µM. I would suggest to include an additional concentration at 1 µM.
Analogously, for AGS cells, to show the results at a concentration of 1 µM would be helpful in the western blots.
Figure 4c and d
Upper data shown in the upper histograms are consistent between colony numbers and and tegaserod failure inhibition of HGC27 and AGS viability.
On the contrary, the histograms at bottom do not show clear effects of tegaserod in cell viability compared to the effect on colony numbers. Please, comment on these differences.
It seems that the effects on MEK2 are lower than those on MEK1.
Figure 6f
The quality of western blot is low. The band corresponding to trametinib lanes of pERK1/2T202/Y204 cannot be interpreted. There are bubbles in lanes 10 and 11 thus the controls cannot be interpreted.
In the results section 3.1, 6th line, the value of IC50, 1,40 microM must be the one for 24h.
What is an advantage to use tegaserod maleate respect to Trametinib ?
Reviewer 2 Report
The manuscript entitled “Tegaserod maleate suppresses the growth of gastric cancer in vivo and in vitro by targeting MEK1/2” provides novel knowledge that may contribute to improve the treatment of gastric cancer using tegaserod maleate as an MEK1/2 inhibitor. I read the manuscript with interest, and I think that the results obtained in this manuscript may be had much interest by many researchers who study the treatment of gastric cancer. However, there are some concerns in this manuscript. Concerns raised are shown as below.
Concerns:
1) Inhibition mechanism of tegaserod maleate.
The authors showed that tegaserod maleate bind with MEK1 at D208, M146 and D152 and MEK2 at G81, G83, respectively. These amino acid residues are located in the kinase domain of these two kinases. Is the inhibition mechanism of tegaserod maleate competitive inhibition? Why can tegaserod maleate bind with kinase domain of these enzymes? In addition, I also expect that you explain the cell permeability of tegaserod maleate in this manuscript.
2) Results 3.4.
The authors described “3.4. Tagaserod maleate inhibits gastric cell growth depend on MEK1/2” in the “Results” section. I think that the authors should change “gastric cell” to “gastric cancer cell”.
3) Western blot analysis.
I think that the authors should carry out quantitative analysis. For instance, in Fig. 4A and 4B, I cannot evaluate the degree of effect of MEK1 and MEK2 gene knockout on the protein levels of MEK1 and MEK2 proteins followed by cell proliferation.
4) Figure 4.
The protein levels of MEK1 or MEK2 did not seem to reflect inhibition rate of proliferation (Fig. 4A and 4B). What do you think about this contradiction? In addition, I think that the authors should describe the efficiency of knockout as not only OD values but also inhibition rate (Fig. 4B) for a better understanding of these experiments.
Did knockout of MEK1 and Mek2 genes cause apoptosis in Fig. 4D?
5) The advantage of tegaserod maleate over trametinib.
The authors insist that there was no statistically significant difference between tegaserod maleate-treated group and trametinib-treated group. Does tegaserod maleate have an advantage over trametinib? How about its side effects, for example cytotoxicity against normal cells including normal gastric cells? I expect that you explain firmly my questions.
